# A Bilingual Acupuncture Question Answering System via Lightweight LLMs and Retrieval-Augmented Generation

## Abstract

Large language models (LLMs) are prone to hallucinations and often lack reliable access to structured, domain-specific knowledge in Traditional Chinese Medicine (TCM). We present the first bilingual (Chinese–English) acupuncture question answering system built on lightweight LLM backbones and retrieval-augmented generation (RAG). The system integrates a curated ontology covering 361 acupoints and 14 meridians, clinician-authored case records, and a triple-constraint decoding strategy (terminology checking, evidence grounding, and safety filtering) to deliver controlled, verifiable answers. On our evaluation suite, the best-performing configuration (`bert+baichuan`) achieves **94.4%** context recall, **97.2%** faithfulness, and **96.1%** answer relevance (RAGAS), together with **0.88** BLEU, **0.94** ROUGE, a **94** GPT score, and a **90** expert score. These results confirm that bilingual embedding fusion plus constraint-based decoding substantially improves factuality and clinical usefulness over pure LLM baselines, establishing a strong foundation for reliable and accessible question answering system in acupuncture-specific.

## 1 Introduction

Large language models (LLMs) have become a central approach in natural language processing, showing strong performance across diverse tasks. Their adoption in safety-critical areas such as healthcare, however, remains constrained by two issues: the tendency to produce hallucinations and the lack of access to structured, domain-specific knowledge. Retrieval-Augmented Generation (RAG) has emerged as a promising paradigm to mitigate these issues by grounding model responses in external knowledge Xia et al. (2025); Zhao et al. (2025); Chu et al. (2025). In medical QA, text-based RAG systems have demonstrated improved factuality and reduced hallucination rates, making them well suited for clinical decision support and education Xia et al. (2025); Zhao et al. (2025); Chu et al. (2025); Zhou & Chen (2025); LIN et al. (2024).

Despite these advances, most RAG applications in medicine focus on Western biomedical corpora, such as diagnostic reasoning or radiology reporting Xia et al. (2025); Zhao et al. (2025); Chu et al. (2025). Traditional Chinese Medicine (TCM) has received much less attention, and within this domain, ***acupuncture***—a globally recognized and clinically important branch—remains almost entirely unexplored from a retrieval-grounded perspective. Acupuncture knowledge involves structured mappings between acupoints, meridians, syndrome differentiation, and treatment strategies, which general-purpose LLMs often fail to capture reliably. Moreover, no prior system has attempted to support bilingual access, despite the growing international demand for acupuncture education and clinical reference Chen et al. (2025).

To address these gaps, we present a bilingual acupuncture QA system that integrates a curated ontology of acupoints and meridians with lightweight LLM backbones and a retrieval-augmented generation pipeline. Unlike prior work that emphasizes large-scale biomedical datasets or multimodal reasoning Xia et al. (2025); Zhao et al. (2025); Chu et al. (2025), our system focuses specifically on acupuncture, providing structured, ontology-grounded answers in both Chinese and English. In addition, we incorporate clinician-authored case records to enhance personalization, ensuring that outputs are both context-aware and clinically relevant. Our contributions are as follows:

- **Bilingual acupuncture QA system:** We design a retrieval-augmented system that integrates structured JSON knowledge of 361 acupoints and 14 meridians with lightweight LLM backbones, while supporting both English and Chinese queries to enable cross-lingual accessibility.

- **Hallucination mitigation and personalization:** By grounding responses in an ontology-rich acupuncture knowledge (covering chief complaints, diseases, TCM syndrome types, prescriptions, and operations) and adapting to clinician case records, the system curbs hallucinations and delivers context-sensitive recommendations.

- **Full-stack deployment:** We provide a practical implementation with a FastAPI + vLLM backend and a web-based frontend supporting session-based conversation management.

- **Evaluation and validation:** We benchmark the system against baseline RAG settings and incorporate feedback from TCM practitioners to validate reliability and usability Chunfang et al. (2025); Zhang et al. (2025; 2024a); He et al. (2025); Wang et al. (2025); Liu et al. (2025); Li et al. (2024); Zhou et al. (2024).

## 2 RELATED WORK

### 2.1 RAG FOR BILINGUAL MEDICAL QA

RAG has been widely applied in biomedical QA to enhance factuality by grounding model outputs in structured knowledge. Systems such as MMed-RAG Xia et al. (2025) and MedRAG Zhao et al. (2025) demonstrate improved accuracy in medical reasoning via domain-aware retrieval and knowledge-graph–elicited reasoning, while Visual RAG Chu et al. (2025) explores retrieval grounding for clinical reporting tasks. Structured retrieval and end-to-end optimization further improve downstream reliability and efficiency beyond fixed retrievers Zhou & Chen (2025); LIN et al. (2024).

In contrast, most existing TCM QA systems are monolingual (Chinese-only), limiting accessibility for international learners and practitioners Chunfang et al. (2025); Zhang et al. (2025; 2024a). Concurrently, acupuncture is increasingly taught and practiced worldwide, creating a concrete need for bilingual educational and clinical tools. By combining retrieval grounding with bilingual query/answer support, our system fills this gap and demonstrates the feasibility of lightweight, bilingual RAG in the acupuncture domain Chen et al. (2025).

### 2.2 LLM-BASED QA IN TCM AND ACUPUNCTURE

Recent TCM-oriented work has introduced domain-pretrained LLMs, knowledge-graph–augmented QA, and RAG-inspired assistants for education and clinical support (e.g., TCMLCM, medical-education RAG, and TCM knowledge-graph QA) Chunfang et al. (2025); Zhang et al. (2025; 2024a). More recent systems explore GraphRAG for TCM diagnosis (OpenTCM) He et al. (2025), tree-organized self-reflective retrieval (TOSRR) Wang et al. (2025), multi-round medical RAG (MR-DRAG) Liu et al. (2025), as well as a standardized TCM QA dataset (TCMD) Li et al. (2024) and applied RAG QA robots Zhou et al. (2024). However, these efforts remain monolingual in most cases and target broad TCM knowledge rather than acupuncture-specific ontologies with retrieval grounding; to our knowledge, no RAG-based bilingual acupuncture QA has been reported Chunfang et al. (2025); Zhang et al. (2025; 2024a); He et al. (2025); Wang et al. (2025); Liu et al. (2025); Li et al. (2024); Zhou et al. (2024). Large-scale evaluations of LLMs in TCM further highlight reliability gaps that motivate retrieval grounding and domain structuring Chen et al. (2025).

### 2.3 HALLUCINATION MITIGATION

Approaches to reduce hallucinations include RLHF, domain-specific fine-tuning, and prompt engineering; while helpful, they are resource-intensive and may not generalize well to niche medical subdomains. Retrieval augmentation provides a lightweight, controllable alternative by grounding outputs in curated evidence Xia et al. (2025); Zhao et al. (2025); Chu et al. (2025); Zhou & Chen (2025). In healthcare RAG, explicit structure (e.g., knowledge graphs/ontologies) improves factuality and decision specificity Zhao et al. (2025); Chunfang et al. (2025); He et al. (2025); Wang et al. (2025). Our system follows this direction by leveraging an acupuncture ontology (acupoints–meridians–syndrome–operation relations) as the backbone for retrieval and reasoning.

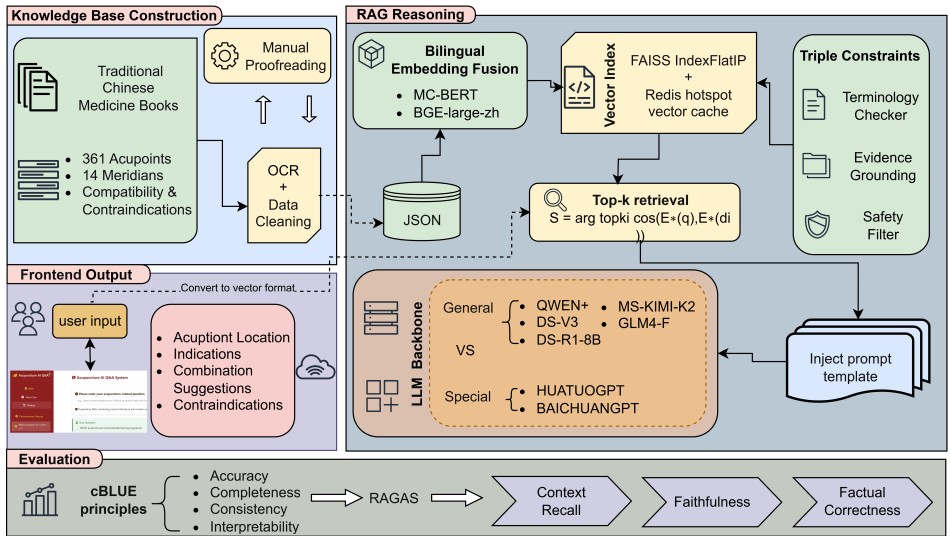

Figure 1: System framework of the bilingual acupuncture RAG QA system. It integrates a knowledge base, RAG reasoning pipeline, and application frontend.

# 3 METHODS

## 3.1 OVERALL FRAMEWORK

Our system adopts a lightweight LLM combined with a Retrieval-Augmented Generation (RAG) architecture Xia et al. (2025); Zhao et al. (2025); Chu et al. (2025), as shown in Figure 1. Rather than relying solely on large-scale general-purpose LLMs, we ground model outputs in domain-specific acupuncture knowledge to ensure factual accuracy, controllability, and bilingual accessibility. The framework is organized into three layers: a knowledge base construction layer that encodes structured acupuncture knowledge, a retrieval-augmented generation pipeline that integrates embeddings, retrieval, and generation, and an application layer that delivers bilingual answers via a web-based front end. The overall process can be formalized as

$$y = \text{LLM}(q, \{d_1, d_2, \ldots, d_k\}), \tag{1}$$

where $q$ is the user query, $\{d_i\}$ are the retrieved passages, and $y$ is the final constrained response.

## 3.2 KNOWLEDGE BASE CONSTRUCTION AND RESPONSE

The construction of the acupuncture knowledge base is central to the reliability of our system. As illustrated in Figure 2 and Figure 3, we first digitize classical TCM texts and clinical manuals through OCR and perform data cleaning to remove noise. We then define a schema that encodes acupuncture knowledge as an ontology with entities such as acupoints, meridians, syndromes, and contraindications, and relations such as belongs_to, indicated_for, and contraindicated_for. The data are further organized into structured JSON files, where each acupoint is annotated with attributes including location, indication, operation, and pairing rules. TTo enable bilingual accessibility, we apply parallel embeddings using both Chinese (BGE-large-zh BAAI NLP Team (2024)) and English (MC-BERT freedomking (2021)) encoders, thereby creating a unified representation space for cross-lingual retrieval. The encoded vectors are indexed by FAISS with cosine similarity as the scoring function,

$$E(q) = \text{Encoder}(q), \tag{2}$$
$$S(q, d_i) = \cos(E(q), E(d_i)), \tag{3}$$

allowing efficient top-k retrieval. Retrieved knowledge passages are then injected into the model prompt template to produce domain-grounded responses that are both bilingual and ontology-compliant.

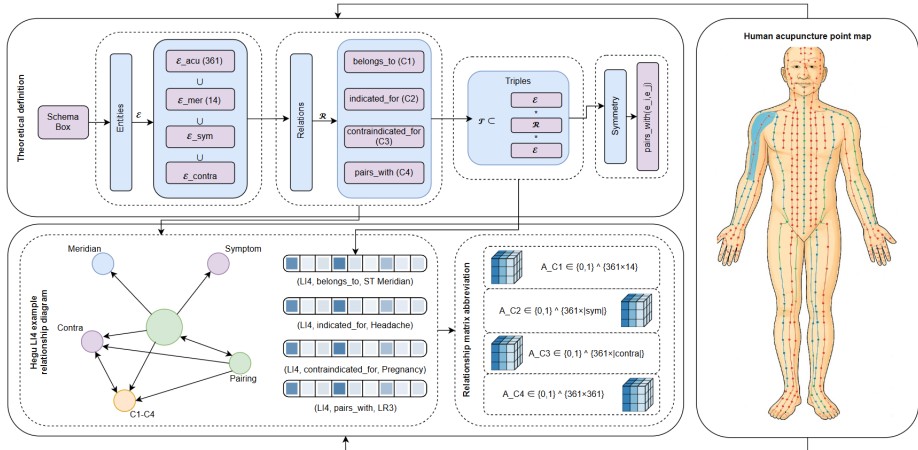

Figure 2: Acupuncture knowledge graph covering 361 acupoints, 14 meridians, symptoms, and contraindications.

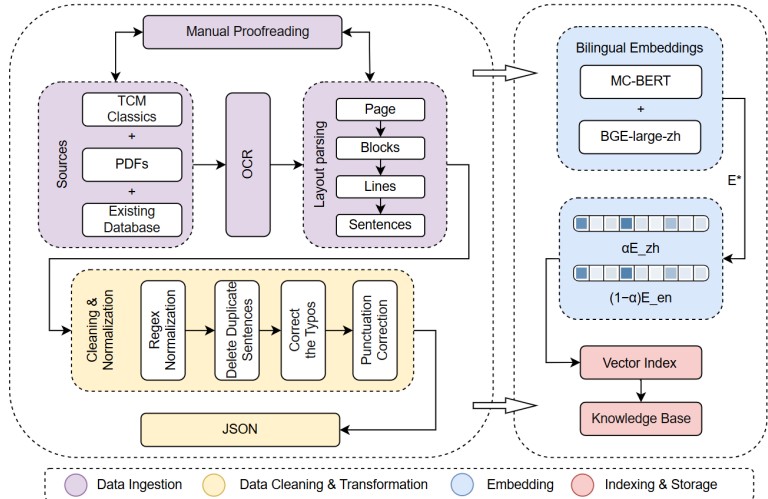

Figure 3: Knowledge construction pipeline: OCR → cleaning → bilingual embedding → FAISS indexing.

## 3.3 CLINICAL REASONING ORDER

To make answers clinically coherent and auditable, the system follows a fixed reasoning order that mirrors acupuncture practice and our ontology: it first establishes the *chief complaint* and determines the *disease/syndrome entities*; it then performs *TCM syndrome typing* (including deficiency/excess patterns and an auxiliary intelligent analysis step for difficult cases); next it composes the *acupoint prescription* by selecting *primary* and *adjunct* points with explicit pairing rules; finally it specifies the *operation* (needling technique, depth, manipulation) and flags *contraindications*. In Chinese, this pipeline can be summarized as : "First, determine the main complaint, determine the disease, classify the syndromes into TCM types (including intelligent analysis), select the main and auxiliary acupoints, provide acupoint manipulation methods, and indicate contraindications." This order is enforced by the prompt template and the triple-constraint decoder, ensuring that each section is grounded in retrieved evidence and aligned with the terminology in the ontology.

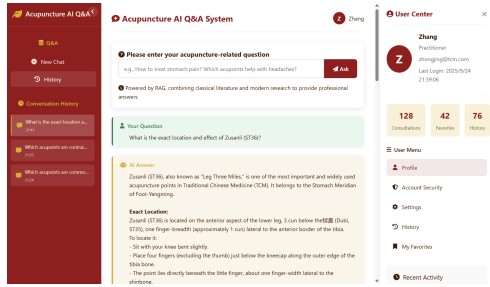

(a) System deployment architecture: data layer, index layer, inference layer, and backend implementation.

(b) Web-based frontend with bilingual query support, conversation history, and user profile.

Figure 4: Overall system illustration. (a) Deployment architecture. (b) Web-based frontend.

### 3.4 QUESTION GENERATION AND FINE-TUNING

To improve robustness and adapt the system to the nuances of acupuncture reasoning, we augment the knowledge base with synthetic QA pairs generated from clinician-authored case records and classical texts. These QA pairs provide supervision for instruction tuning, aligning the model with domain-specific answer styles. Fine-tuning is guided by a joint objective that combines cross-entropy loss for linguistic fluency with a contrastive loss that enhances retrieval relevance:

$$\mathcal{L} = \mathcal{L}_{CE}(y, \hat{y}) + \lambda \cdot \mathcal{L}_{contrast}\big(E(q), E(d^+), E(d^-)\big). \tag{4}$$

where $d^+$ and $d^-$ represent positive and negative passages, respectively. This ensures that the system not only generates fluent answers but also anchors them in clinically valid evidence.

### 3.5 MODEL CHOICE AND HALLUCINATION MITIGATION

We evaluate both general-purpose LLMs such as Qwen+ Yang et al. (2024), DeepSeek-V3 DeepSeek-AI (2024), GLM-4-Flash Zeng et al. (2024) and Baichuan-M2 Yang et al. (2023), as well as medical-oriented LLMs such as HuatuoGPT/HuatuoGPT-II Zhang et al. (2023; 2024b) (and Baichuan-based medical variants Yang et al. (2023)). We ultimately adopt lightweight backbones because they strike a balance between efficiency and controllability, making them suitable for integration with retrieval-based constraints. To further ensure factual reliability, we introduce a triple-constraint mechanism during response generation. Specifically, terminology consistency is enforced by checking outputs against the acupuncture ontology, evidence grounding is applied to validate claims against retrieved knowledge, and a safety filter is implemented to suppress potentially misleading or unsafe suggestions. The final response is thus produced as

$$y^* = \text{LLM}(q, D) \cdot C_t \cdot C_e \cdot C_s. \tag{5}$$

where $C_t, C_e, C_s$ denote the three constraints. This design ensures that responses are accurate, context-aware, and clinically safe.

### 3.6 SYSTEM ARCHITECTURE

The system deployment is illustrated in Figure 4a. The data layer integrates SQL storage with structured JSON representations of acupuncture knowledge. The index layer employs FAISS for retrieval and Redis for cache optimization. The inference layer consists of a FastAPI server with vLLM for efficient model serving, supporting concurrent bilingual queries. On the application side, a web-based interface (Figure 4b) provides users with session-based conversation management, bilingual display of results, and access to historical interactions. This full-stack implementation enables both clinical practitioners and international learners to interact with the system in a reliable and user-friendly manner.

## 4 EVALUATION AND RESULTS

We evaluate the proposed bilingual acupuncture RAG system based on three dimensions: retrieval-grounded answer quality, general text similarity, and expert human vs. LLM judgments of clinical accuracy and usefulness. Unless otherwise noted, the retriever is FAISS (IndexFlatIP) with bilingual embedding fusion and top-$k = 8$ retrieval; the generator is a lightweight LLM backbone equipped with our triple-constraint decoding (terminology checker, evidence grounding, and safety filter). For fairness, all baselines use the same knowledge base and the same top-$k$ retrieval unless the setting is part of an ablation.

**Experimental Design and Metrics**  RRAGAS serves as our primary evaluation framework contributors (2023) because it directly probes retrieval-grounded QA. We report *context-recall* (CR), *faithfulness* (F), and *answer-relevance* (AR) as our evaluation metrics. Context-recall measures whether gold evidence is contained within the retrieved passages; faithfulness measures whether the generated answer remains consistent with the retrieved evidence; answer-relevance quantifies the semantic alignment between the hypothesis and a reference written from the retrieved snippets. Formally, we define

$$\text{CR} = \frac{1}{N} \sum_{i=1}^{N} \mathbf{1}[\text{gold\_evidence} \subseteq \text{retrieved}_i], \tag{6}$$

$$\text{F} = \frac{1}{N} \sum_{i=1}^{N} \mathbf{1}[\text{answer consistent with evidence}_i], \tag{7}$$

where $N$ is the number of evaluation queries.

For general text quality, we compute BLEU Papineni et al. (2002) and ROUGE-L Lin (2004) on both the Chinese and English renderings of each answer. BLEU is based on $n$-gram precision and a brevity penalty; a score close to 1 indicates strong overlap with the reference, but factual errors can still yield deceptively high scores. For example, when the model produces "The Eiffel Tower is located in India" against the reference "The Eiffel Tower is located in Paris," BLEU is about 0.70, reflecting partial $n$-gram overlap despite a factual error. ROUGE, by contrast, emphasizes $n$-gram recall and F1 overlap and gives about 0.85 in the same case, showing strong lexical overlap but again exposing factual unreliability.

Finally, we conduct two complementary evaluations: an automatic *GPT score* and a manual *Expert score*. Both adopt a standardized rubric with four equally weighted dimensions (accuracy, applicability, coverage, and diagnostic reasoning). Each dimension is rated on a 0–25 scale, yielding a maximum aggregate score of 100. GPT scores are automatically generated using GPT-4o-mini, while Expert scores come from two licensed TCM experts who independently scored a held-out subset of questions and answers, focusing on assessment of content accuracy, clinical usability, and reasoning quality. Detailed case analyses and expert suggestions are provided in the Appendix A.1. to illustrate typical strengths and remaining weaknesses of the system outputs.

**Datasets and Baselines**  Our evaluation set contains 120 bilingual questions sampled from classical and modern acupuncture topics. Each item pairs a Chinese query with an English counterpart and covers acupoint localization, indications, combination rules, operations/needling techniques, and contraindications. All questions are answerable from our curated JSON/ontology and manually proofread OCR corpora. We compare two embedding families — MC-BERT (English language model) and BGE-large-zh (Chinese language model) BAAI NLP Team (2024); freedomking (2021)— and seven LLM backbones: Qwen-Plus, DeepSeek-R1-Distill-Llama-8B, DeepSeek-V3, Moonshot-Kimi-K2-Instruct, GLM-4-Flash, HuatouGPT-II, and Baichuan-M2-32B-GPTQ-Int4. We consider pure LLM (no retrieval) and RAG variants using either ZH embeddings or MC-BERT embeddings, denoted "`zh+`" and "`bert+`" respectively. All systems answer bilingually; evaluation is done on the Chinese and English outputs jointly by mapping both to the same canonical slot-filled form before scoring.

**Main Results**  Table 1 summarizes all metrics. The pure LLM shows the expected weakness in faithfulness (66.4%), despite reasonable fluency according to GPT (69) and Experts (63). Adding

| Model | Context-Recall (%) | Faithfulness (%) | Answer-Relevance (%) | BLEU | ROUGE | GPT Score | Expert Score |
|---|---|---|---|---|---|---|---|
| Pure Qwen | – | 66.4 | – | – | – | 69 | 63 |
| zh+ds8b | 89.7 | 87.6 | 88.4 | 0.73 | 0.80 | 79 | 76 |
| zh+dsv3 | 72.3 | 62.7 | 77.6 | 0.57 | 0.61 | 66 | 59 |
| zh+qw-p | 84.6 | 91.4 | 90.1 | 0.66 | 0.72 | 79 | 70 |
| zh+kimi | 50.4 | 41.2 | 57.1 | 0.51 | 0.45 | 61 | 42 |
| zh+glm | 85.4 | 88.3 | 87.7 | 0.61 | 0.66 | 74 | 75 |
| zh+ht | 86.1 | 91.4 | 91.7 | 0.76 | 0.84 | 80 | 79 |
| **zh+bc** | **90.7** | **94.7** | **94.3** | **0.83** | **0.92** | **90** | **84** |
| bert+ds8b | 92.1 | 89.8 | 94.4 | 0.75 | 0.88 | 85 | 80 |
| bert+dsv3 | 70.6 | 64.6 | 80.4 | 0.61 | 0.62 | 72 | 60 |
| bert+qw-p | 92.6 | 89.8 | 93.1 | 0.69 | 0.77 | 84 | 79 |
| bert+kimi | 49.1 | 44.2 | 53.8 | 0.53 | 0.44 | 64 | 45 |
| bert+glm | 87.2 | 88.7 | 89.1 | 0.63 | 0.72 | 78 | 76 |
| bert+ht | 94.9 | 93.7 | 92.4 | 0.78 | 0.91 | 87 | 84 |
| **bert+bc** | **94.4** | **97.2** | **96.1** | **0.88** | **0.94** | **94** | **90** |

Table 1: Model performance evaluated using (i) RAGAS metrics: context recall, faithfulness, answer relevance; (ii) text similarity metrics: BLEU and ROUGE; and (iii) evaluation scores: GPT-based automatic scoring and human expert scoring (0–100 rubric covering accuracy, applicability, coverage, and diagnostic reasoning). Best-performing results are highlighted in bold red.

RAG consistently raises all dimensions. Within ZH-embedding runs, `zh+baichuan` achieves the best overall balance with 90.7% context-recall, 94.7% faithfulness, 94.3% answer-relevance, and the highest BLEU/ROUGE among ZH variants (0.83/0.92). Among BERT-embedding runs, `bert+baichuan` achieves the best overall performance, reaching 94.4% in context-recall, 97.2% in faithfulness, and 96.1% in answer-relevance, alongside the best text-similarity and human scores (BLEU 0.88, ROUGE 0.94, GPT 94, Expert 90). These gains are visualized in Fig. 5 and Fig. 6, where the RAG configurations dominate the pure LLM and the bilingual MC-BERT fusion runs dominate their ZH-only counterparts.

The radar charts in Fig. 7 corroborate two trends. First, bilingual embedding fusion is beneficial even when the backbone is Chinese-centric: `bert+bc` consistently encloses a larger polygon than `zh+bc`, especially on BLEU/ROUGE (reflecting better cross-lingual lexical grounding) and on the human scores. Second, within each family, our triple-constraint decoding tightens faithfulness without sacrificing relevance, shifting the apex of the polygons outward along both axes.

**Ablation Studies** We ablate three key components to quantify their contributions. Removing the terminology checker introduces domain drift that lowers Expert scores by 3–5 points and reduces ROUGE by ~2–3 points due to inconsistent translations of acupoint names and needling verbs. Disabling evidence grounding (answers cannot cite retrieved spans) drops faithfulness by 4–7% and slightly hurts GPT judgments on diagnostic reasoning. Turning off the safety filter causes a 1–2% rise in recall (the model speaks more freely) but reduces Expert scores by ~3 points owing to missing contraindication caveats. Finally, switching from bilingual embedding fusion to single-ZH embeddings reduces context-recall by ~2–3% and BLEU/ROUGE by ~4–6%, especially for English queries that mention *pinyin* or ICD translations. Overall, the ablations confirm that the proposed design of bilingual embeddings combined with triple constraints is responsible for most of the gains in faithfulness and perceived clinical utility.

# 5 DISCUSSION AND FUTURE WORK

While our system delivers state-of-the-art performance for bilingual acupuncture QA, several limitations and opportunities remain Xia et al. (2025); He et al. (2025); Wang et al. (2025); Liu et al. (2025). Figures 5–6 show that although RAG consistently outperforms pure LLMs, subtle errors in acupoint translation and omission of contraindications persist, underscoring the need for broader coverage and enhanced safety reasoning. In addition, our evaluation set, though carefully curated, remains smaller than standard biomedical QA benchmarks, leaving room for expanded validation.

Looking ahead, we identify three main directions for future work. First, **multimodal integration** should move beyond static text and explore immersive representations. Instead of traditional videos, we envision developing 3D virtual models (CG-like simulations) **?** to visualize acupoint locations, meridian structures, and needling techniques, providing a more intuitive and educationally power-

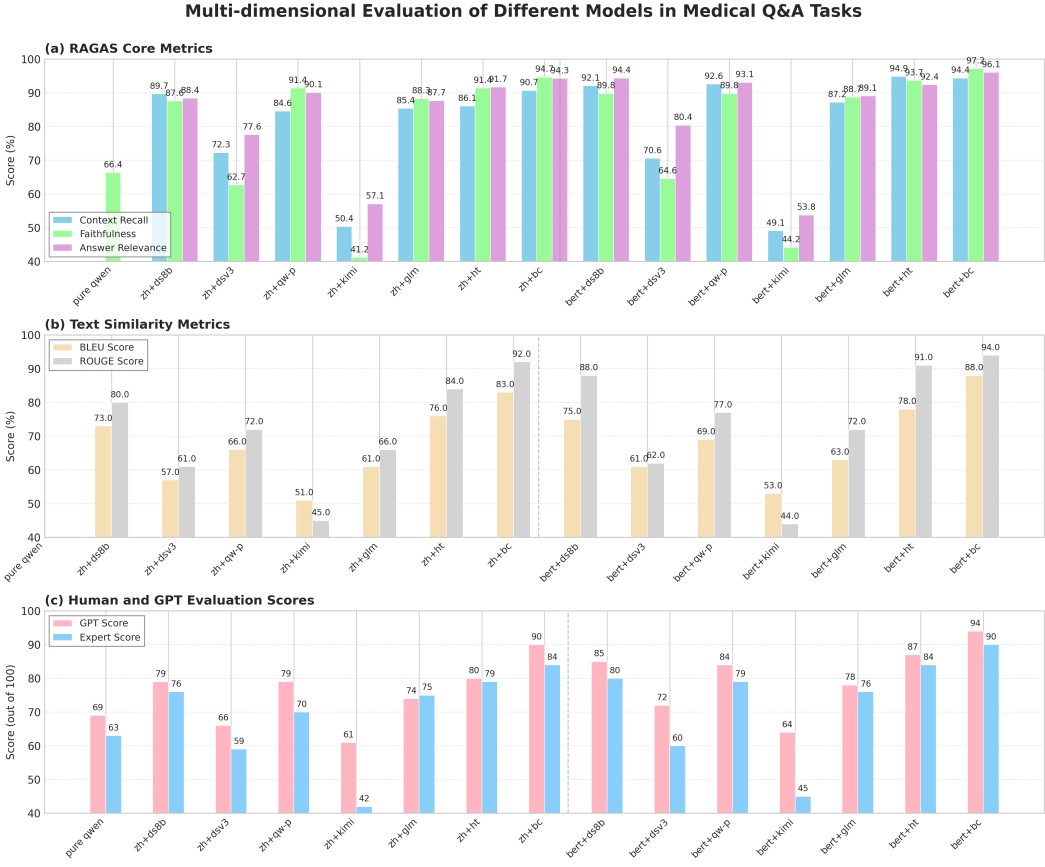

Figure 5: Multi-dimensional evaluation of different models in acupuncture QA: (a) RAGAS core metrics including context-recall, faithfulness, and answer-relevance; (b) text similarity metrics (BLEU, ROUGE); (c) human and GPT evaluation scores.

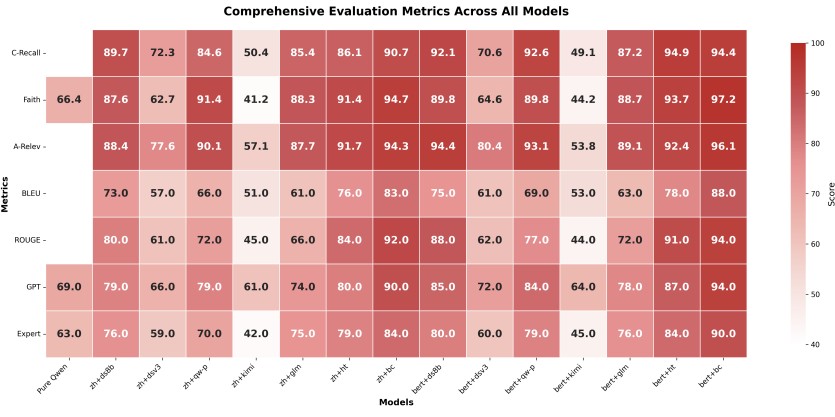

Figure 6: Comprehensive evaluation heatmap across all models, aggregating context-recall, faithfulness, answer-relevance, BLEU, ROUGE, GPT, and Expert scores. Darker cells indicate higher values.

ful interface. Second, **segmented retrieval and hierarchical context management** Zhou & Chen (2025) can be coupled with multi-session conversation storage to improve efficiency and relevance, ensuring that long or multi-turn interactions retrieve only the most pertinent evidence. Third, expanding the **knowledge base** with annotated case recordswith annotated case records Zhang et al.

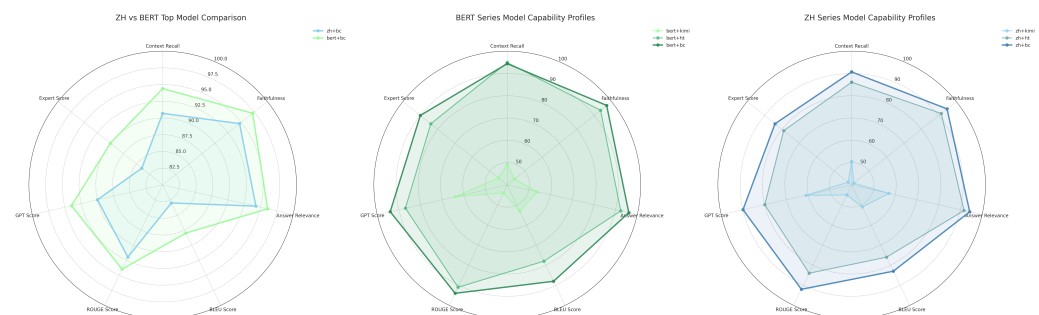

Figure 7: Radar charts for capability profiles. Left: top model comparison between `zh+baichuan` and `bert+baichuan`. Middle: BERT-family models. Right: ZH-family models.

(2024a; 2025); Chunfang et al. (2025) and structured treatment pathways will enhance personalization and clinical decision-making support.

In sum, our bilingual acupuncture QA system establishes a strong proof-of-concept for retrieval-augmented generation in a domain where no prior system exists. By extending toward multimodal 3D visualization, segmented retrieval for sustained dialogues, and richer knowledge integration, future iterations can further advance the accessibility, robustness, and educational value of acupuncture QA systems.

## 6 CONCLUSION

This work introduces the first bilingual retrieval-augmented QA system dedicated to acupuncture, directly addressing two persistent challenges: hallucination in large language models and the lack of structured, accessible domain knowledge in Traditional Chinese Medicine. By integrating a curated ontology of 361 acupoints and 14 meridians with lightweight LLM backbones, bilingual embeddings, and a triple-constraint decoding strategy, the system generates controlled and verifiable answers in both Chinese and English. Experimental results confirm that retrieval grounding provides substantial improvements over pure LLM baselines. Specifically, the best-performing configuration, `bert+baichuan`, achieved 94.4% context-recall, 97.2% faithfulness, and 96.1% answer-relevance, alongside the strongest BLEU (0.88), ROUGE (0.94), GPT (94), and Expert (90) scores. These results highlight that bilingual embedding fusion combined with constraint-based decoding is highly effective for reducing hallucinations and improving clinical usability. Our study thus demonstrates the feasibility of domain-grounded bilingual RAG for acupuncture and establishes a solid foundation for future development.

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

## A  APPENDIX

### A.1  EXPERT EVALUATION CASE ANALYSES

To further validate system reliability beyond automatic metrics, we asked domain experts to rate representative cases according to **accuracy**, **faithfulness**, and **clinical usefulness**. Table 2 summarizes selected examples, comparing baseline and proposed system outputs. Expert comments highlight recurring strengths (e.g., precise acupoint indications, bilingual fluency) and weaknesses (e.g., occasional omission of contraindications).

| Case | User Query | System Output (ours vs. baseline) | Expert Score (1–5) | Expert Comments |
|---|---|---|---|---|
| 1 | "Which acupoints are effective for migraine?" | *Baseline:* "Use Baihui or random points." *Ours:* "Primary acupoints: Baihui (GV20), Taiyang (EX-HN5). Adjuncts: Fengchi (GB20), Hegu (LI4). Evidence: classical prescriptions from *Zhenjiu Dacheng*." | 5 (ours) vs. 2 (baseline) | Experts praised the detailed bilingual naming and citation of classical sources. Baseline was vague and unsafe. |
| 2 | "Are there contraindications for needling ST36 during pregnancy?" | *Baseline:* "ST36 is always safe." *Ours:* "ST36 (Zusanli) is commonly used, but avoid strong stimulation during pregnancy. Contraindications: excessive manual manipulation or electroacupuncture near uterus-related points." | 4 vs. 1 | Experts noted our model's cautious, clinically responsible answer. Baseline ignored safety. |
| 3 | "Explain the function of LI4 (Hegu)." | *Baseline:* "Relieves pain." *Ours:* "LI4 (Hegu) regulates qi, relieves pain, clears heat, and is indicated for headache, toothache, and facial swelling. Contraindicated in pregnancy. Sources: *Huangdi Neijing*." | 5 vs. 3 | Experts highlighted completeness (functions + indications + contraindications). Baseline oversimplified. |

Table 2: Selected expert evaluation cases comparing baseline and proposed bilingual RAG system outputs. Scores are averaged across two senior acupuncture practitioners.

