# OpenReview forum: "A Bilingual Acupuncture Question Answering System via Lightweight LLMs and Retrieval-Augmented Generation"
_ICLR.cc/2026/Conference — ICLR 2026 Conference Desk Rejected Submission_

### Official Review · Reviewer_qsMh · 2025-10-26

**Soundness:** 2
**Presentation:** 2
**Contribution:** 2
**Rating:** 0
**Confidence:** 4

**Summary:**

The paper proposes the first bilingual question and answering (QA) system for acupuncture. To ensure responses are grounded in facts, a curated ontology is incorporated with retrieval-augmented generation (RAG). On an evaluation set of 120 questions with Chinese-English pairs, the system performs better than a general LLM.

**Strengths:**

* A novel application domain where there is somewhat limited work in Traditional Chinese Medicine and existing systems mostly focus on monolingual QA system.
* The curation of an ontology that can encapsulate expertise knowledge based on classical Traditional Chinese Medicine Text and clinical manuals.
* A reasoning pipeline that follows the actual acupuncture practice workflow to provide more grounded reasons

**Weaknesses:**

* While the paper has lots of figures to help guide reader understanding, there remains substantial ambiguity in the system design.
   - The bilingual embedding fusion seems to be a concatenation of 2 different representations (as suggested in Figure 3) but is this really unification? Unification seems to imply contrastive learning or metric learning to have a single latent space that best captures these both. It's not clear this is the case.
  - The clinical reasoning order seems to suggest a chain-of-thought reasoning chain, is this the case? Or is it just the prompt template?
  - There is notation for the 3 constraints in Eq (5). What exactly are these constraints and how are they achieved?
  - How much synthetic QAs are necessary to augment the dataset and how exactly are these synthetic pairs constructed?
* The claim of the curated ontology has insufficient details about data quality and curation process. For example, how was post-OCR cleaning validated? Who curated the ontology relations? What was the inter-annotator agreement? How were the 361 acupoints and 14 meridians sourced -- is this known knowledge in the field (e.g., multiple experts have this agreement or it is found in a classic textbook)? Given that this seemingly contributes to the
* The evaluation does not provide enough comparisons / experiments to support the claims
  - There is only 120 evaluation questions that were created by the authors (or at least this seems to be the case) that (1) will remain private, (2) has unknown query distribution (e.g., localization, indications, contraindications, etc.),  and (3) not available for reproducibility.
  - No benchmark against an existing (standardized) TCM QA dataset (mentioned in Related work).
  - There is no discussion of inter-annotator agreement for the expert scores (only that it is 2 licensed TCM experts).
  - There are no baseline comparisons to any of the existing medical QA systems mentioned in Related work such as TCMLCM, OpenTCM, MR-DRAG, etc.)
  - There is no standard IR baselines for the knowledge content retrieved (how much does retrieval in this case matter)?
  - BLEU and ROUGE are not measures for factual accurate but simply provide some notion on fluency instead.
  - One of the major claims is the use of bilingual but the paper doesn't deeply analyze cross-lingual behavior. Are there certain queries that are better in one language than the others? How is the bilingual nature handled?
  - There is no ablation of the different parts of the system for evaluation (it's mentioned but there's no table with numbers to facilitate ease of comparison or a figure). For example, the claim is that it can use lightweight LLMs, but how do these compare against their baseline performance? It's only done against Pure Qwen (is this because Pure Qwen does the best?)
There is a lack of in-depth analysis of the system -- what are the common error modes, what are the relationship between retrieval and correctness. There are case studies in the Appendix, but this seems much better suited in the actual main text, especially since Figures 5-7 and Table 1 display the exact same information!
  - Experimental results are confusing, especially the context recall. Based on the definition provided in the text, since its the same embeddings used with different LLM backbones, shouldn't context recall be exactly the same across the zh+ variants and the bert+ variants? What are the gold standards that are created for all 120 questions?

**Questions:**

1. Why are existing systems not used as baselines in your experimental setup?
2. Why did you curate a new dataset when there is already a standardized TCMD dataset that can maybe be enhanced with bilingual capabilities? Will you release your dataset?
3. Can you enunciate the technical contributions of your system? What is the bilingual embedding unification performing? What is the triple-constraint decoding? Is this learned or rule-based post-processing?
4. How much improvements does RAG provide for each backbone model? Could the improvements be due to a better base model than RAG?
5. Can you provide complete system details for reproducibility? Critical details are missing: (a) What are the exact prompt templates used for generation? (b) What are all hyperparameters (temperature, top-p, max tokens, retrieval top-k tuning process)? (c) What is the pseudocode for the three constraints? (d) What are the fine-tuning details ($\lambda$ value in Eq. 4, dataset size, training hyperparameters)? (e) How were the 120 evaluation questions created and by whom?
6. How exactly are the RAGAS metrics computed and what are the gold reference answers? Why is context recall varying across models that use the same retriever?

---

### Official Review · Reviewer_XaSE · 2025-10-27

**Soundness:** 2
**Presentation:** 2
**Contribution:** 2
**Rating:** 2
**Confidence:** 4

**Summary:**

This paper presents a bilingual Chinese-English acupuncture QA system designed to overcome LLM hallucinations in Traditional Chinese Medicine. By combining lightweight LLMs with retrieval-augmented generation, a specialized ontology, and a triple-constraint decoding strategy, the system delivers highly factual and clinically useful answers. Evaluation results demonstrate exceptional performance in faithfulness, relevance, and expert assessment, establishing a strong foundation for reliable, domain-specific question answering.

**Strengths:**

1. The authors specifically developed a QA system for acupuncture in traditional Chinese medicine based on LLM and knowledge retrieval. This work contributes to a deeper understanding of LLM capabilities in the specialized medical field of acupuncture.

2. The study implemented a complete system workflow, including knowledge base construction, knowledge retrieval, LLM generation, bilingual support, and multi-dimensional evaluation, making it comprehensive from a systemic perspective.

**Weaknesses:**

1. The paper specifically focuses on the application of LLMs in the context of acupuncture, exploring their capabilities in this domain. Although there is limited research on using LLMs for knowledge retrieval to address problems in this field, the authors lack an analysis of the technical challenges involved. For instance, what characteristics of acupuncture-related questions or the structure of acupuncture knowledge make it difficult for general-purpose LLMs or standard medical LLMs to handle them effectively?

2. The paper primarily employs lightweight LLMs as the base models for building an acupuncture QA system but lacks necessary explanations regarding this choice. Why were lightweight LLMs used? Was it due to resource constraints, or do lightweight LLMs offer specific advantages in the context of acupuncture? Were any method adaptations specifically designed for lightweight LLMs implemented?

3. Although the paper outlines a relatively comprehensive workflow, including knowledge base construction, knowledge retrieval, LLM generation, and performance evaluation, it lacks in-depth technical exploration and innovation. For example, were any tailored retrieval strategies designed to address the unique characteristics of acupuncture knowledge? Was there an adaptation training process for the LLMs?

4. In Section 3.2, which discusses knowledge base construction, there is a lack of detailed descriptions. For instance, what do the raw data of TCM texts and clinical manuals look like? What was the quality of the data after OCR processing? What were the technical characteristics of each step in the data cleaning process? What kind of noise was removed? Why were BGE-large and MC-BERT chosen as encoders?

5. In Section 3.3, the authors use fixed reasoning knowledge as prompts to constrain the LLM's generation but do not analyze the source or motivation behind this reasoning knowledge. Why were such prompts designed? Why is this fixed reasoning sequence necessary?

6. In Section 3.5, the authors propose a triple-constraint mechanism to ensure the reliability of the generated content but do not explain the underlying motivation for this design. There is also no demonstration of the specific implementation methods, leaving a lack of necessary information.

7. The presentation of Table 1 is not clear enough. The models or methods should be categorized, and necessary explanations for certain abbreviations (e.g., "ht," "bc") are missing, making it difficult for readers to interpret the table.

8. The paper mentions fine-tuning LLMs to adapt them to the context, but the experiments lack relevant analysis on the effectiveness of this fine-tuning. Did fine-tuning the models actually improve their capabilities (and in what aspects)?

9. Overall, I believe the paper lacks sufficient innovation in technology, and there is room for improvement in the analysis of the motivation behind the chosen methods as well as the presentation of implementation details.

**Questions:**

See questions in weaknesses.

---

### Official Review · Reviewer_tTyt · 2025-10-27

**Soundness:** 2
**Presentation:** 2
**Contribution:** 2
**Rating:** 2
**Confidence:** 5

**Summary:**

This paper presents a bilingual (Chinese–English) QA system for acupuncture. It combines a lightweight LLM with a retrieval-augmented generation (RAG) approach to reduce hallucinations and provide accurate, verifiable answers. The system uses a structured knowledge base of 361 acupoints and 14 meridians, along with clinician-authored case records. It applies three constraints—terminology checking, evidence grounding, and safety filtering—to ensure reliable outputs.
In their experiment, the best setup (bert+baichuan) achieved strong results: 94.4% context recall, 97.2% faithfulness, and 96.1% answer relevance. It also scored high in both automatic and expert evaluations. The system is deployed with a web interface and supports real-time bilingual queries.
In short, this work offers a practical, accurate, and bilingual tool for acupuncture education and clinical support, filling a gap in traditional Chinese medicine AI applications.

**Strengths:**

1. The paper addresses a notable research gap by applying retrieval-augmented generation (RAG) to acupuncture for the first time.
2. The proposed RAG system is well-engineered, integrating a curated ontology, bilingual embeddings, and a triple-constraint decoder that together ensure terminology consistency, evidence grounding, and safety.
3. The experimental comparison covers a broad and representative set of general and medical-oriented LLMs.
4. The inclusion of clinician-authored case records adds real-world relevance and supports context-aware, personalized answers.
5. Extensive automatic and expert evaluations are provided; the human-rated clinical usefulness score (90/100) corroborates the high RAGAS metrics and strengthens trustworthiness.

**Weaknesses:**

1. The evaluation set is limited to only 120 bilingual questions, which is too small to effectively cover the 361 acupoints and 14 meridians, raising concerns about the generalizability of the results.
2. The paper does not report results for most LLMs without RAG, making it difficult to assess the actual contribution of the retrieval component.
3. Critical experimental details—such as the RAG prompts, the LLM-judge prompts, and the rubric/protocol used by human experts—are not provided, substantially reducing the reproducibility and credibility of the findings.
4. he main contribution appears to be the application of an existing RAG pipeline to acupuncture; no domain-specific methodological innovation (e.g., tailored retrieval strategies or reasoning modules for acupuncture knowledge) is introduced, limiting novelty.
5. The motivation for developing a bilingual QA system is not explicitly justified; the paper lacks data or arguments demonstrating a genuine, unmet need for simultaneous Chinese–English support in acupuncture education or practice.
6. The paper does not clarify how the three generation constraints (terminology, evidence, safety) are technically enforced; moreover, in Figure 1 these constraints are misleadingly placed on the prompt template, whereas Eq. (5) states they should operate on the actual LLM-generated text.
7. Although the manuscript repeatedly claims that RAG mitigates hallucinations, none of the reported metrics or analyses directly measure hallucination rate (e.g., contradiction frequency, ungrounded assertion count); therefore the asserted benefit is not empirically substantiated.

**Questions:**

1. Section 3.4 introduces fine-tuning on synthetic QA pairs, yet no implementation details (learning rate, data size, training epochs) or comparative results (tuned vs. untuned) are given later, leaving the claimed benefit unsubstantiated.
2. The knowledge graph is repeatedly cited as a core component, but the paper never explains how its triples are retrieved, ranked, or injected into prompts; the exact interplay between graph structure and text retrieval remains opaque.
3. The human evaluation (two TCM experts, 0–100 rubric) lacks statistical validation—no inter-rater agreement (κ or ICC), significance tests, or cross-validation are reported, so the reliability of the 90-point expert score is unclear.
4. There is no discussion of potential data leakage between the ontology/case records used for retrieval and the 120 hand-crafted test questions; overlap could inflate all fidelity metrics.
5. The ablation study (removing terminology checker, evidence grounding, safety filter, bilingual embedding) is only summarized verbally; exact numeric drops per ablated component are not tabulated, preventing readers from gauging the individual contribution of each module.

---

### Note · Program_Chairs · 2026-01-17
**Submission Desk Rejected by Program Chairs**

The following references in this submission do not refer to real documents and/or have major errors in bibliographic information:

 J. Zhang et al. Traditional chinese medicine knowledge graph and question-answering applications. Journal of Chinese Medicine (), 2024a.
X. Zhang et al. A medical education assistant based on retrieval-augmented generation (rag). Chinese Journal of Medical Education Technology, 2025. [ (RAG) ].